REGISTERED REPORT PROTOCOL

# Effect of accelerated postoperative rehabilitation after tibial tubercle distalisation: A randomised controlled trial protocol

Timo Rahnel[1]*, Frederick K. Weitz[2], Ville M. Mattila[3], Aleksi Reito[3], Erkki Nilkku[4], Antti P. Launonen[3], Petri J. Sillanpää[2]

1 Department of Orthopaedic Surgery, North Estonia Medical Centre, Tallinn, Estonia, 2 Pihlajalinna, Koskisairaala Hospital, Tampere, Finland, 3 Department of Orthopaedic Surgery, Tampere University Hospital, Tampere, Finland, 4 Department of Physiotherapy, Pihlajalinna, Koskiklinikka, Tampere, Finland

☯ These authors contributed equally to this work.
* timo.rahnel@regionaalhaigla.ee

## Abstract

Patella alta is a clinical condition where the patella is positioned too proximal in relation to the femoral trochlea. Such an abnormality may cause patellar instability and predispose to recurrent patellofemoral dislocations and patellofemoral pain. There are no conclusive guidelines for determining a threshold for too high positioned patella, as several different methods have been described to measure patellar height. As a surgical solution, distalising tibial tubercle osteotomy has been described to correct excessive patellar height. In the early phase of the distalising tibial tubercle osteotomy postoperative protocol, weightbearing and knee flexion are limited with a brace commonly for 4–8 weeks to avoid potential implant failure leading to displacement of the osteotomy or non-union. The potential risks for adverse effects associated with the limitation rehabilitation protocol include a delay in regaining knee range of motion, stiffness and muscle weakness. As a result, recovery from surgery is delayed and may lead to additional procedures and long-term morbidity in knee function. This is a prospective, randomised, controlled, single-blinded, single centre trial comparing a novel accelerated rehabilitation protocol with the traditional, motion restricting rehabilitation protocol. All skeletally mature patients aged 35 years and younger, referred to as the distalising tibial tubercle osteotomy procedure group, are eligible for inclusion in the study. Patients will be randomised to either the fast rehabilitation group or the traditional rehabilitation group. Patients with patellar instability will be additionally treated with medial patellofemoral ligament reconstruction. The hypothesis of the trial is that the novel accelerated rehabilitation protocol will lead to faster recovery and improved functional outcome at 6, 12 and 24 weeks compared with the conservative rehabilitation protocol. A secondary hypothesis is that the complication rate will be similar in both groups. The study will document short-term recovery and the planned follow-up will be 3 years. After the 1-year follow-up, the trial results will be disseminated in a major peer-reviewed orthopaedic publication. Protocol version 3.6, date 28/11/2023.

**Data Availability Statement:** All relevant data from this study will be made available upon study completion. The access rules are not defined yet.

**Funding:** The authors received no specific funding for this work.

**Competing interests:** The authors have declared that no competing interests exist.

## Introduction

The term patella alta refers to the abnormal height of the patellar, where the anatomical position of the patellar is too proximal. In cases of patella alta, patellar and trochlear cartilage surfaces have limited or no contact. This anatomical variation can cause patellar instability and result in recurrent patellofemoral dislocations. One of the possible symptoms caused by patella alta can be patellofemoral pain with or without patellar instability. In addition, patella alta is a significant risk-factor for patellofemoral dislocations and anterior knee pain.

Several methods have been described to measure patellar height in the scientific literature [1]: the most commonly used methods as the Insall-Salvati index [2], the Blackburne-Peel index and the Caton-Deschamps index [3, 4]. The patellotrochlear index (PTI) is a novel measure to concretise the contact area between the patellar and trochlear joint surfaces [1].

Distalising tibial tubercle osteotomy (DTTO) is a suitable surgical procedure to treat patella alta. [5, 6]. During the early rehabilitation phase after DTTO, it is common to limit weight-bearing and knee flexion with orthosis. In general, the orthosis is used for between 4 and 8 weeks and limited weightbearing is maintained for 4 to 6 weeks to avoid potential implant failure leading to displacement of the osteotomy or non-union. The potential risks of this rehabilitation protocol are restricted knee range of motion (ROM) and muscle weakness, leading to delayed recovery and the possible need for additional procedures.

In our previous, as yet unpublished, retrospective study, we have shown that accelerated rehabilitation is a feasible treatment method after DTTO, and we believe that patient aftercare can be more aggressive than at present. However, to the best of our knowledge, no high-quality randomised controlled trials have been conducted to date on postoperative rehabilitation protocols after DTTO.

Thus, the aim of this randomised controlled trial is to identify the differences between the results of two postoperative rehabilitation protocols. One of the protocols is a typical conservative rehabilitation protocol and other is a novel accelerated rehabilitation protocol.

## Materials and methods

In this is prospective, randomised, controlled, single-blinded, single centre trial, we compare an accelerated rehabilitation protocol with a typical conservative rehabilitation protocol after DTTO (Fig 1 and Table 1). The trial will be reported in accordance with the Consolidated Standards of Reporting Trials (CONSORT) guidelines.

The hypothesis of the trial is that the novel accelerated rehabilitation protocol will lead to faster recovery and improved functional outcome at 6, 12 and 24 weeks compared with the conservative rehabilitation protocol. The secondary hypothesis is that the complication rate will be similar for both protocols.

The primary outcome in this study will be knee ROM measured at 12 weeks postoperatively. A total difference regardless of the direction of the movement of 10° in full range of movement will be considered significant. The results will be measured with a long goniometer in a standardised manner. Secondary outcomes will be knee ROM measured at baseline, 6, 24 and 52 weeks postoperatively, the Knee Injury and Osteoarthritis Outcome Score (KOOS) and the Banff Patella Instability Instrument (BPII) score measured at baseline, 6, 12, 24 and 52 weeks, and isometric muscle strength measured at baseline, 12, 24 and 52 weeks postoperatively. Regained isometric muscle strength will be measured during follow-up visits and calculated to a ratio compared with a preoperative value. The number of reoperations and complications (failure of fixation and stress-fractures) will be also reported as secondary outcomes. The KOOS questionnaire is an instrument to assess patients' opinions about their

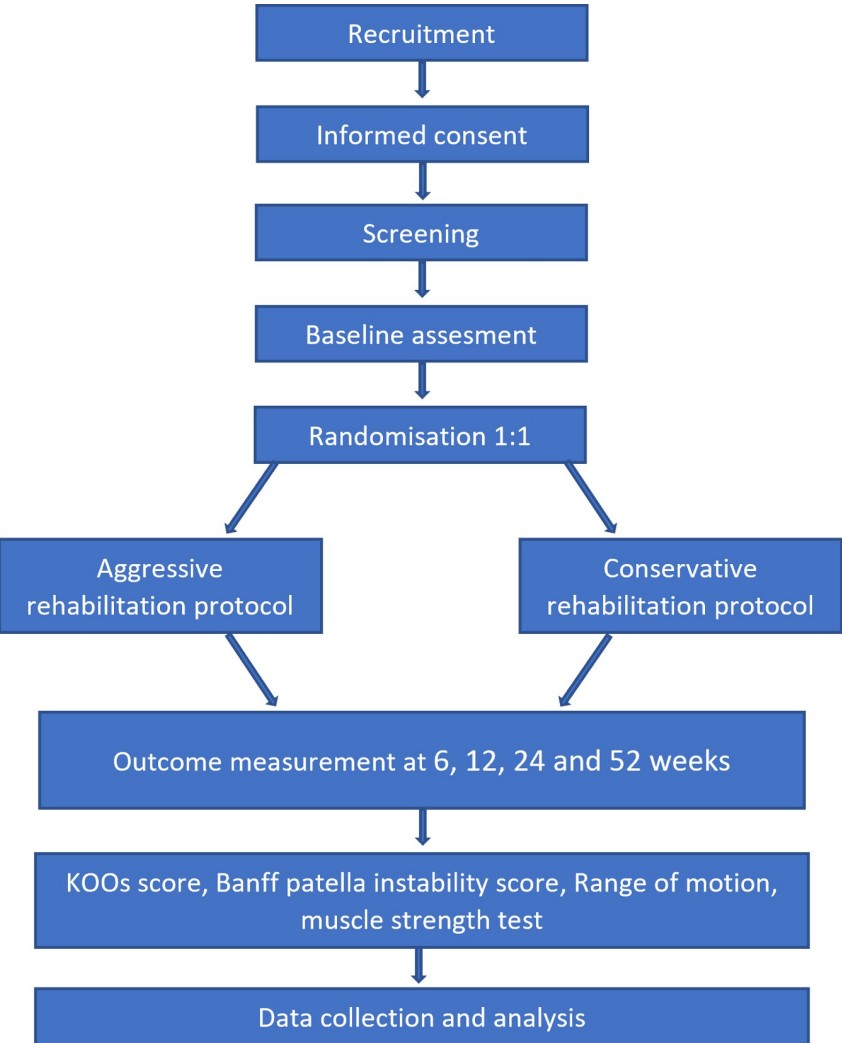

**Fig 1. Study flow chart.**

knees and any associated problems. KOOS consists of 5 subscales: Pain, other Symptoms, Function in daily living, Function in sport and recreation and knee related Quality of life. BPII is also a patient-reported outcome measurement instrument. BPII is, however, more specific for patellofemoral problems.

After enrolment, all the patients visit the study physiotherapist. The baseline scores from ROM, isometric muscle strength, KOOS and BPII are recorded during the visit. Randomisation to the allocation groups will be done after surgery.

The ROM measurements are taken with a goniometer, with the patient in supine position. The active ROM is measured as the patient moves the knee in a full range of active flexion and extension. ROM is then measured. Passive ROM is measured in the same supine position. The examiner first moves the knee in a full range of flexion and then in a full range of extension. ROM is then measured. Full ROM is 5º-0º-140º. A ROM of 0º-130º is considered adequate.

The knee extension isometric power measurement is performed with the participant in the sitting position. The isometric power is measured with belts-stabilized dynamometer placed in

**Table 1. Study design and assessments.**

| | Study period | | | | | |
|---|---|---|---|---|---|---|
| | Enrolment | Allocation | Post allocation | | | |
| Study visit | 1 | 2 | 3 | 4 | 5 | 6 |
| Timepoint | Preoperative | Surgery day (0) | 6 weeks | 12 weeks | 24 weeks | 52 weeks |
| **Enrolment** | | | | | | |
| Eligibility screening | x | | | | | |
| Informed consent | x | | | | | |
| Demographics and medical data | x | | | | | |
| Allocation | | x | | | | |
| **Intervention** | | | | | | |
| TT-distalisation | | x | | | | |
| **Assessments** | | | | | | |
| BPII score | x | | x | x | x | x |
| KOOS | x | | x | x | x | x |
| Adverse events | | x | x | x | x | x |

TT-distalisation–distalisation of tibial tubercle; BPII–Banff Patella Instability Instrument; KOOS–Knee Injury and Osteoarthritis Outcome Score.

the support pilar of the treatment table and connected to participants ankle with anchor belt placed 5 cm above the ankle.

The measurement is taken with a dynamometer at three different knee angles 5˚, 30˚ and 90˚. KOOS and BPI are taken before the muscle strength and ROM measurements.

Ethical approval has been obtained from the local health authority. Patients recruited and those patients who declined to take part in the trial but joined the follow-up cohort will provide written informed consent. All data will be deidentified and the results will be published at the group level only. Individual patients will not be identifiable.

The trial is registered in Tampere University Hospital medical research ethics committee (toimikunta.eettinen@pshp.fi) on 15.11.2022 with registration number R22072.The trial is registered in ClinicalTrials.gov, Identifier: NCT05854056

## Patient selection

The eligible study population will comprise all consecutive patients aged 15 to 35 years with closed growth plate (physis) who have been referred for DTTO surgery at Pihlajalinna Koski-sairaala Hospital, Tampere, Finland. The upper age limit was chosen to minimize the effect of patellofemoral osteoarthritis. As the DTTO procedure can only be planned for skeletally mature patients, the closure of the physis will be evaluated with magnetic resonance imaging (MRI) and plain radiographs. The Greulich and Pyle method will be used to define skeletal age if needed [7].

Only those patients with isolated DTTO or DTTO combined with medial patellofemoral ligament reconstruction will be included in the study. At the initial out-patient ward the treating physician will give the information about the study and ask for written informed consent.

The following criteria will be used for patient selection throughout the study.

❖ Inclusion criteria

• Symptomatic patella alta with recurrent dislocation or subluxation. Long-lasting anterior knee pain not responding to rehabilitation.

❖ Exclusion criteria

➢ Radiographic:

- Open growth plates

- Iwano [8] grade III and IV changes in patellofemoral joint

- Caton-Deschamps <1.2 in MRI

- PTI >30% in MRI

- High grade trochlear dysplasia

➢ General:

- Refuses to participate in the study

- Aged less than 15 or more than 35 years

- Severe neurological, pulmonal or cardiovascular comorbidities that are contraindications for surgery

- Lack of adequate co-operation

- Does not adequately understand written and spoken instructions in the local language

Those patients who decline to take part in the trial will be asked to join a follow-up cohort as a background population. The patients in this follow-up cohort will be treated "as normal" without allocation, but the follow-up questionnaires will be the same as those given to the randomly assigned population. The patients in the follow-up cohort will also be asked to provide informed written consent.

## Randomisation

Patients will be randomised using a pre-trial random number matrix in block allocation fashion in varying block. We have two stratification factors. First is age (under 20 and 20–35 years) since age has been shown to associate with the main outcome measure. Another stratification factor will be history of the patellar dislocation i.e. group 1- history with at least 1 documented patellar dislocation and group 2-history with no documented patellar dislocation.

Randomisation will be performed using an online randomisation platform (Redcap). After informed written consent has been received and the intervention in the OR performed, the patients will be randomly allocated for rehabilitation. The physician responsible for the intervention or treatment will not participate in the collection of patient outcomes during the follow-up. Patients are encouraged not to reveal the group to the assessor, so assessor will be blinded. The Patients research data will be concealed to Redcap program until the blinded analysis and two versions of discussions has been agreed. After which the group will be revealed. The research coordinator will monitor the study flow. An independent monitoring committee, which was established during our previous randomised controlled trial, will monitor the study.

## Surgical treatment

Operative treatment will be performed by trained and experienced knee surgeons. The surgeons' skills and number of procedures will be reported according to the criteria given by the Consort Group [9].

The aim of the surgical treatment is to normalise the biomechanics of the patellofemoral joint. The operation will start with an arthroscopic examination to verify patella alta position. The high-positioned patella in relation to the trochlea will be documented and an approximation made of the required distalisation for adequate cartilage contact between the patella and the trochlea. Thereafter, a 5–7 cm mid-line incision will be made at the tibial tubercle. The gracilis graft will be harvested from the same incision if needed for medial patellofemoral ligament reconstruction. The lateral aspect of the tibial tubercle will be exposed. The proximal distalisation cut will be made in AP direction at 70º angle at a minimum of 6–7 cm distally from the tibial tubercle. The distal AP cut will be done according to preoperative measurements taken from imagining data and the arthroscopic findings of the patient. The bone block will be removed to allow distalisation of the tubercle. Furthermore, the tubercle will be mobilised with chisels to allow the distalisation of the tubercle. The distalisation is at least 5 mm, usually even more. The osteotomy will be fixed with two 4.5 mm AO screws. The appropriate localisation of the fixation and patella will be confirmed with fluoroscopy.

## Rehabilitation

Patients will be randomised in two rehabilitation groups. Patients in both groups will be guided by in-ward physiotherapists and will be given written physiotherapy guidelines for both instructed physiotherapy and self-guided exercises. After discharge from the hospital, patients will be referred to physiotherapy for further guidance. Both groups will start preliminary exercises from the first postoperative day to reduce oedema in the operated lower limb. For the detailed rehabilitation guidelines, please see Table 2.

## Follow-up

Postoperative physiotherapy control visits will take place at 6, 12, 24 and 52 weeks after the procedure. The physiotherapy control visits will be managed by a senior physiotherapist who will be blinded to the allocation group. During the visits, knee ROM and isometric flexion/extension strength will be measured. The isometric extension and flexion strength will be measured with an SBS-KW-300SLIM scale. In addition, the PROMs (KOOS and BPI) will be collected during the research related control visits by the senior physiotherapist at previously defined time points (Table 2).

Postoperative physiotherapy sessions will be provided by physiotherapists who will not otherwise be involved in the study. Physiotherapy control visits and rehabilitation guidance sessions with the treating physiotherapist will take place at 1, 4, 6, 8, 12 and 24 weeks. The

**Table 2. The rehabilitation guidelines.**

| Elements of physical therapy | Group 1 Accelerated rehabilitation | Group 2 Conservative rehabilitation |
|---|---|---|
| Antioedema knee, calf, leg | Day 1 | Day 1 |
| Knee brace, ROM limitations | No | 6 weeks |
| Weight-bearing limitations | No, crutches are recommended for 4 weeks. | 4 weeks limb weight, followed by half body weight till week 6 |
| Active exercises. Functional exercises. | 1 week | 8 weeks |
| Active dynamic strengthening exercises. | 5 weeks | 8 weeks |
| Closed chain exercises for muscle strengthening. | 8 weeks | 8 weeks |
| Muscle endurance and neuromuscular control, progress strengthening exercises, jogging | 12 weeks | 12 weeks |

physiotherapy protocol will last for 24 weeks. Protocols 1 and 2 are presented in the supplementary material.

Healing of the osteotomy site will be evaluated with plain radiograph examination at 6 and 12 weeks after the procedure. The radiographs will be evaluated by an experienced radiologist and the operating surgeons. Physician visits will be at 6 and 12 weeks. Knee scores and isometric flexion/extension strength will be assessed preoperatively and at 12 and 52 weeks postoperatively.

## Power analysis

We defined 10 degrees [10] in knee ROM in the primary end-point as a clinically relevant difference we would not like miss. We estimated a common standard population deviation of 20 degrees for knee ROM in both groups. Based on unadjusted t-test (TrialSize package in R), we need 63 patients per group to have a power of 80% with 5% type I error level. With this age group, the estimate of loss to follow-up rate will be set to 5% and will result in a total of 132 patients participating in the trial.

## Statistical analysis

Study analysis will be done using mixed linear model allowing for repeated measures in longitudinal setting. All continuous variables will be analyses in this fashion. Each outcome variable will be the dependent variable and patient the random factor. Study group and time point will be included as fixed factor. Stratification variables are included as covariates in the analysis. If baseline value for the dependent variable (i.e. ROM) is available, it will be included also as a covariate. Treatment effects at each time point are the estimated marginal means for time point and study group interaction term. Satterthwaite method is used to estimate degrees in freedom in these analyses. No imputation is needed as liner mixed model is used and a full data set will be used in the analyses. Logistic regression is used to analyse binary outcome. Adjusted risk difference at each time point will be reported for complications and reoperations. All analyses will done with RStudio with suitable packages (i.e. *lmer* and *emmean*).

## Data management plan

Electronic portfolios provided by the Redcap application at Tampere university hospital's server will be used to collect patient research data. Patients will be de-identified: each patient will be assigned a unique trial identification number (TIN), which is then matched with the patient's personal identification number. The identification of each patient will only be possible after retrieving the matching key, which will be stored in a locked locker in the research nurse's office at Pihlajalinna Koskisairaala Hospital. Access to patient research data will not be given to parties outside the trial. Throughout the trial, the research data will only be handled with a TIN.

Data will be saved electronically by patients and research personnel to Tampere University Hospital's secure research server–Redcap (via tablets and laptops), which has been security cleared by the hospital district. The research data saved to the server will contain only pseudonymous TINs with a set of numbers acquired from the questionnaires; that is, each question is answered with a number. This will ensure the anonymity of each individual patient, and that the identity of the patient will remain secret, even if server data are revealed to third parties.

Each researcher participating in the trial will gain access to the data at the end of the trial for further analyses. All variables in the data set will be described, and suitable metadata standards will be used, when available.

The copyright of the trial research data will be owned and created by the participating research parties. The data will be shared among all participating researchers who will receive access to the data after the trial is completed. Due to confidentiality and legal agreements, public data sharing will be restricted because we only have permission to hold the data in the specific research server, not to transfer data. Under certain circumstances, for example, when a new member joins the collaboration, we will grant access to the data.

Members of study group have access to the password-protected study data for analysis. All research team members are bound by professional secrecy with respect to identifiable patient information. The results of the study will be published in a form that does not identify those individuals who participated in the study.

## Interim analysis

The external trial board will execute the interim analysis after half of the patients have been recruited. The analysis will focus on the number of adverse events and the trial board, based on the results of the analysis, will give a recommendation as to whether the trial should continue. If complications rate in accelerated rehabilitation group is remarkably increased in that phase, the study is discarded. Adverse events and serious adverse events will be reported according to the recommendations given by the Consort Group. They will be reported as complications in the final analysis.

Adverse event is defined as follows:

➢ Superficial infection

➢ Persisting pain, complex regional pain syndrome or other reasons that can be cured on an ambulatory basis.

Serious adverse event is defined as follows:

➢ Deep infection: Any infection with known bacterial source that calls for re-operation

➢ Implant failure leading to displacement of the osteotomy

➢ Non-union at 12 weeks control visit

## Trial schedule

Approval for the study was sought in the autumn of 2022. Patient recruitment will start in the spring of 2024. The study will document short-term recovery and the planned follow-up will be 1 year. The number of operations will be collected until the spring of 2025. Follow-up control visits will take place until the spring of 2026.

After the 1-year follow-up, the results of the trial will be disseminated in a major peer-reviewed orthopaedic publication.

## Supporting information

**S1 File. Protocol 1.** Accelerated Rehabilitation Program.
(PDF)

**S2 File. Protocol 2.** Conservative Rehabilitation Program.
(PDF)

**S3 File. PT_1.** Personal Exercise Program 1.
(PDF)

**S4 File. PT_2.** Personal Exercise Program 2.
(PDF)

**S5 File. PT_3.** Personal Exercise Program 3.
(PDF)

**S6 File. PT_4.** Personal Exercise Program 4.
(PDF)

**S7 File. PT_5.** Personal Exercise Program 5.
(PDF)

**S8 File. PT_6.** Personal Exercise Program 6.
(PDF)

**S9 File. SPIRIT checklist.**
(DOC)

**S10 File. Trial study protocol, original language.**
(DOCX)

**S11 File. Trial study protocol, translation.**
(DOCX)

## Author Contributions

**Conceptualization:** Frederick K. Weitz, Antti P. Launonen, Petri J. Sillanpää.

**Data curation:** Frederick K. Weitz, Erkki Nilkku, Antti P. Launonen, Petri J. Sillanpää.

**Formal analysis:** Ville M. Mattila, Aleksi Reito, Antti P. Launonen, Petri J. Sillanpää.

**Investigation:** Erkki Nilkku.

**Methodology:** Ville M. Mattila, Aleksi Reito, Antti P. Launonen, Petri J. Sillanpää.

**Project administration:** Timo Rahnel, Antti P. Launonen.

**Software:** Antti P. Launonen.

**Supervision:** Ville M. Mattila, Aleksi Reito, Antti P. Launonen, Petri J. Sillanpää.

**Validation:** Aleksi Reito, Antti P. Launonen.

**Visualization:** Timo Rahnel.

**Writing – original draft:** Timo Rahnel, Frederick K. Weitz, Aleksi Reito.

**Writing – review & editing:** Timo Rahnel, Frederick K. Weitz, Aleksi Reito, Erkki Nilkku, Antti P. Launonen, Petri J. Sillanpää.

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
