## [Decision Letter · Decision Letter 0]

10 Jul 2023

PONE-D-23-09669Effect of accelerated postoperative rehabilitation after tibial tubercle distalisation: a randomised controlled trial protocolPLOS ONE

Dear Dr. Rahnel,

Thank you for submitting your manuscript to PLOS ONE. After careful consideration, we feel that it has merit but does not fully meet PLOS ONE’s publication criteria as it currently stands. Therefore, we invite you to submit a revised version of the manuscript that addresses the points raised during the review process.

We look forward to receiving your revised manuscript.

Kind regards,

Ahmed A. Khalifa, M.D., FRCS, MSc.

Academic Editor

PLOS ONE

Journal Requirements:

Reviewers' comments:

Reviewer's Responses to Questions

**Comments to the Author**

1. Does the manuscript provide a valid rationale for the proposed study, with clearly identified and justified research questions?

Reviewer #1: Yes

Reviewer #2: Partly

2. Is the protocol technically sound and planned in a manner that will lead to a meaningful outcome and allow testing the stated hypotheses?

Reviewer #1: Yes

Reviewer #2: Partly

3. Is the methodology feasible and described in sufficient detail to allow the work to be replicable?

Reviewer #1: Yes

Reviewer #2: Yes

4. Have the authors described where all data underlying the findings will be made available when the study is complete?

Reviewer #1: No

Reviewer #2: Yes

5. Is the manuscript presented in an intelligible fashion and written in standard English?

Reviewer #1: Yes

Reviewer #2: Yes

6. Review Comments to the Author

You may also provide optional suggestions and comments to authors that they might find helpful in planning their study.

Reviewer #1: The authors reported about the protocol of a prospective, randomised, controlled, single-blinded, single centre trial to evaluate the effect of accelerated postoperative with the traditional rehabilitation after tibial tubercle distalisation with respect to knee ROM measured at 12 weeks.

In general the current protocol is sound and - based on the fact the recruitment is not yet started (according to clintrial.gov) the minor comments can be corrected in the protocol without affecting validity.

Detailed comments:

L95: I think it should be "will be" instead of "has been".

L121: I understand the RO measurement in the supine position is the main outcome measure. However, it is not clear, if the difference between the treatment groups is in flexation direction, or extension direction or the range (of flexation to extension). This should be made clear in the paper.

L166: The stratification /randomization is not clear. As the total sample size 144 falls in two treatment groups and stratified by 2 age groups this calls for 72 per treatment group which is not divisible by 10 the block size. The authors also mentioned, that " Caton-Deschamps index" is another stratification factor. However strata categories are not given. I assume that, according the inclusion criteria, no correspond subgroups are considered. I would not recommend to use blocksizes of 8 to avoid unfilled blocks, but rather to use blocksizes of varying length (6,8,10) or BigStick randomization to mitigate attrition bias.

Please describe how concealment is achieved. Please comment, how detection bias - do to single blinding - is achieved, e.g. by keeping patients records concealed.

L223: It is acceptable to use an unadjusted (unstratified) test as basis for sample size calculation. Please give the name of the test and the name of the statistical software used for calculation. I assume it is a t-Test with equal variances in both groups. (My software tells me 64 per group).

Concerning the dropout rate: The portion of 15% is large and is prone to invalidate study results. If the authors use a multiple imputation (MI) analysis model based on the longitudinal data and the corresponding aggregated analysis for the t-Test, no correction for missing is necessary. This would protect against a under/overpowered study, if the observed missing rate is different from the planned.

L229: The statistical analysis model has to be modified. The primary analysis is a stratified t-Test (Age) mit MI aggregation for missing. The sensitivity analysis may include a mixed model for Repeated measures (MMRM, Mallinckrod et al (2013), "Recent Developments in the Prevention and Treatment of Missing Data")) using the longitudinal observations. The latter should be used for the secondary outcomes as well.

pleas give the statistical software planned to do the analysis.

Reviewer #2: I believe the research question is valid and clinically relevant

Line 29–30

I believe the authors should mention the reasons for holding knee motion and limiting weight-bearing status somewhere in their preamble

Line 34

I’m not sure that I would call their accelerated rehab program ‘novel’ as many surgeons already do it.

However it is never been studied in a prospective randomized way, which is the uniqueness of the study.

Line 40–43

Hypothesis is valid and understandable

Line 71 to 78

I believe the surgery to reduce patella alta, and our measuring techniques does not need to be reviewed for the study, and could be considerably shortened.

Line 80–84

I will repeat what was said above. I believe that the rational why some surgeons limit knee motion and weight-bearing status merits at least one line of explanation. The authors go on to discuss the potential risk of the “traditional” rehab program, but they do not talk about the potential risk of an accelerated program.

The primary outcome of a study is knee range of motion. I would not expect this to be different and perhaps should not be the primary outcome, but perhaps moved to a secondary outcome.

A secondary outcome is isometric muscle strength. I believe this is a valid outcome, I would put it as a primary outcome.

A third outcome might be time to bone healing, but that might be difficult to distinguish healing time as they are looking at six week intervals.

Line 1 26–1 27

I’m assuming isometric muscle strength is measured at the tibial level as they’re measuring it at different angles of knee motion. This is an open chain test, specific details of where the dynometer is placed should be given i.e. at the proximal tibia, or the lower distal tibia. If I understand the timing of interventions, this will be tested at six weeks with a maximum isometric contraction, which might be difficult to perform maximally at six weeks due to pain, placement of the dynometer, and or risk of torque at the fracture site, depending on where exactly they’re placing the dynometer.

Line 1 40–1 42: inclusion criteria

If I am reading this correctly as they are treating two different patient populations.

One patient group is the more traditional reason for this surgical intervention, (ie) recurrent patellar dislocation.

The second would be anterior knee pain combined with patella alta. This second category of patients is quite a bit smaller in most of our populations, and I am not sure that it adds to the study to combine the reason for the patient seeking surgery. Also no reason for this treatment for this class of patients in given.

Exclusion criteria line 1 47–1 48

Their surgical thresholds are quite low in my opinion, and this would lead to a very small mm (distance) of distalization.

One value that I do not see mentioned in there results would be to factor in the millimeters of displacement, as I think that 2-3mm of distal repostitioning is different than 8-10 mm. This might pose a different element into their outcomes. Since already the surgical threshold is Lower than many Patellofemoral surgeons, I believe some of their patients will be distalized only a few millimeters, and therefore with a more accelerated program, we have would expect fewer complications.

Line to 12–2 22

Complications

And lines 272 to 278

Adverse events

These two sections seem duplicative to me

7. PLOS authors have the option to publish the peer review history of their article (what does this mean?). If published, this will include your full peer review and any attached files.

Reviewer #1: No

Reviewer #2: No

---

## [Author Response · Author response to Decision Letter 0]

29 Nov 2023

6. Review Comments to the Author

You may also provide optional suggestions and comments to authors that they might find helpful in planning their study.

Reviewer #1: The authors reported about the protocol of a prospective, randomised, controlled, single-blinded, single centre trial to evaluate the effect of accelerated postoperative with the traditional rehabilitation after tibial tubercle distalisation with respect to knee ROM measured at 12 weeks.

In general the current protocol is sound and - based on the fact the recruitment is not yet started (according to clintrial.gov) the minor comments can be corrected in the protocol without affecting validity.

Detailed comments:

L95: I think it should be "will be" instead of "has been".

Corrected, see L92. 

L121: I understand the RO measurement in the supine position is the main outcome measure. However, it is not clear, if the difference between the treatment groups is in flexation direction, or extension direction or the range (of flexation to extension). This should be made clear in the paper.

Corrected, see L102-103.

L166: The stratification /randomization is not clear. As the total sample size 144 falls in two treatment groups and stratified by 2 age groups this calls for 72 per treatment group which is not divisible by 10 the block size. The authors also mentioned, that " Caton-Deschamps index" is another stratification factor. However strata categories are not given. I assume that, according the inclusion criteria, no correspond subgroups are considered. I would not recommend to use blocksizes of 8 to avoid unfilled blocks, but rather to use blocksizes of varying length (6,8,10) or BigStick randomization to mitigate attrition bias.

Please describe how concealment is achieved. Please comment, how detection bias - do to single blinding - is achieved, e.g. by keeping patients records concealed.

Corrected, we use varying block sizes. See L172-173.

Caton-Deschamps index is not used for stratification anymore. 

We added another stratification factor: history of the patellar dislocation i.e. group 1- history with at least 1 documented patellar dislocation and group 2-history with no documented patellar dislocation. See L174-176.

Concealment described, see L 180-183.

L223: It is acceptable to use an unadjusted (unstratified) test as basis for sample size calculation. Please give the name of the test and the name of the statistical software used for calculation. I assume it is a t-Test with equal variances in both groups. (My software tells me 64 per group).

Concerning the dropout rate: The portion of 15% is large and is prone to invalidate study results. If the authors use a multiple imputation (MI) analysis model based on the longitudinal data and the corresponding aggregated analysis for the t-Test, no correction for missing is necessary. This would protect against a under/overpowered study, if the observed missing rate is different from the planned.

Corrected, see L 228-233.

L229: The statistical analysis model has to be modified. The primary analysis is a stratified t-Test (Age) mit MI aggregation for missing. The sensitivity analysis may include a mixed model for Repeated measures (MMRM, Mallinckrod et al (2013), "Recent Developments in the Prevention and Treatment of Missing Data")) using the longitudinal observations. The latter should be used for the secondary outcomes as well.

pleas give the statistical software planned to do the analysis.

Modified, see L 235-245.

Reviewer #2: I believe the research question is valid and clinically relevant

Line 29–30

I believe the authors should mention the reasons for holding knee motion and limiting weight-bearing status somewhere in their preamble

Corrected, see L30-31. 

Line 34

I’m not sure that I would call their accelerated rehab program ‘novel’ as many surgeons already do it.

However it is never been studied in a prospective randomized way, which is the uniqueness of the study.

We agree. 

Line 40–43

Hypothesis is valid and understandable

Line 71 to 78

I believe the surgery to reduce patella alta, and our measuring techniques does not need to be reviewed for the study, and could be considerably shortened.

Corrected, see L71-74.

Line 80–84

I will repeat what was said above. I believe that the rational why some surgeons limit knee motion and weight-bearing status merits at least one line of explanation. The authors go on to discuss the potential risk of the “traditional” rehab program, but they do not talk about the potential risk of an accelerated program.

Corrected, see L30-31 and 78-79.

The primary outcome of a study is knee range of motion. I would not expect this to be different and perhaps should not be the primary outcome, but perhaps moved to a secondary outcome.

A secondary outcome is isometric muscle strength. I believe this is a valid outcome, I would put it as a primary outcome.

A third outcome might be time to bone healing, but that might be difficult to distinguish healing time as they are looking at six week intervals.

We discussed it with our study group and we are on the opinion to keep the primary and secondary outcomes unchanged as we assume that range of motion compared to isometric muscle strength suffers even more with conservative rehabilitation and is a critical risk after tibial tubercle distalisation. Aim is to avoid any limitation in range of motion that could affect everyday life. 

We agree about the possible third outcome. 

Line 1 26–1 27

I’m assuming isometric muscle strength is measured at the tibial level as they’re measuring it at different angles of knee motion. This is an open chain test, specific details of where the dynometer is placed should be given i.e. at the proximal tibia, or the lower distal tibia. If I understand the timing of interventions, this will be tested at six weeks with a maximum isometric contraction, which might be difficult to perform maximally at six weeks due to pain, placement of the dynometer, and or risk of torque at the fracture site, depending on where exactly they’re placing the dynometer.

Corrected, the measuring technique is explained more thoroughly. See L124-127. 

We agree, the timing of the interventions was difficult to distinguish. The maximum isometric contraction will be tested at twelve weeks. See addition on L225-226.

Line 1 40–1 42: inclusion criteria

If I am reading this correctly as they are treating two different patient populations.

One patient group is the more traditional reason for this surgical intervention, (ie) recurrent patellar dislocation.

The second would be anterior knee pain combined with patella alta. This second category of patients is quite a bit smaller in most of our populations, and I am not sure that it adds to the study to combine the reason for the patient seeking surgery. Also no reason for this treatment for this class of patients in given.

We agree that two different patient populations could rise questions in some circumstances but the study’s principle aim is to compare to different rehabilitation protocols after the operative treatment rather than indications for surgery. We believe that in our selected patient group the indication for the surgery doesn´t play role in rehabilitation results. 

As the patient group without any documented patellar dislocation is remarkable, we would like to include these patients to the study as well. That is why we added new stratification factor: history of the patellar dislocation i.e. group 1- history with at least 1 documented patellar dislocation and group 2-history with no documented patellar dislocation. See L174-176.

Exclusion criteria line 1 47–1 48

Their surgical thresholds are quite low in my opinion, and this would lead to a very small mm (distance) of distalization.

One value that I do not see mentioned in there results would be to factor in the millimeters of displacement, as I think that 2-3mm of distal repostitioning is different than 8-10 mm. This might pose a different element into their outcomes. Since already the surgical threshold is Lower than many Patellofemoral surgeons, I believe some of their patients will be distalized only a few millimeters, and therefore with a more accelerated program, we have would expect fewer complications.

Corrected and we totally agree that the distalisation distance great difference will affect the outcomes. 

We increased the surgical thresholds (Caton-Deschamps <1.2 in MRI; PTI >30% in MRI) to avoid the doubts about possible minimal distance of distalisation. See L156-157.

As explanation we distalise always at least 5 mm and usually the value is even bigger. Added to the text, see L200-201. 

Line to 12–2 22

Complications

And lines 272 to 278

Adverse events

These two sections seem duplicative to me

Corrected, see L278-287

7. PLOS authors have the option to publish the peer review history of their article (what does this mean?). If published, this will include your full peer review and any attached files.

Do you want your identity to be public for this peer review? For information about this choice, including consent withdrawal, please see our Privacy Policy.

Reviewer #1: No

Reviewer #2: No

---

## [Decision Letter · Decision Letter 1]

24 Mar 2024

PONE-D-23-09669R1Effect of accelerated postoperative rehabilitation after tibial tubercle distalisation: a randomised controlled trial protocolPLOS ONE

Dear Dr. Rahnel,

Thank you for submitting your manuscript to PLOS ONE. After careful consideration, we feel that it has merit but does not fully meet PLOS ONE’s publication criteria as it currently stands. Therefore, we invite you to submit a revised version of the manuscript that addresses the points raised during the review process.

We look forward to receiving your revised manuscript.

Kind regards,

Mehrnaz Kajbafvala, Ph.D

Academic Editor

PLOS ONE

Journal Requirements:

Reviewers' comments:

Reviewer's Responses to Questions

**Comments to the Author**

1. Does the manuscript provide a valid rationale for the proposed study, with clearly identified and justified research questions?

Reviewer #1: No

Reviewer #3: Yes

2. Is the protocol technically sound and planned in a manner that will lead to a meaningful outcome and allow testing the stated hypotheses?

Reviewer #1: No

Reviewer #3: Yes

3. Is the methodology feasible and described in sufficient detail to allow the work to be replicable?

Reviewer #1: No

Reviewer #3: Yes

4. Have the authors described where all data underlying the findings will be made available when the study is complete?

Reviewer #1: No

Reviewer #3: Yes

5. Is the manuscript presented in an intelligible fashion and written in standard English?

Reviewer #1: Yes

Reviewer #3: Yes

6. Review Comments to the Author

You may also provide optional suggestions and comments to authors that they might find helpful in planning their study.

Reviewer #1: The presented description is not aligned with the specific study, i.e. what method to prove what! Statsistical analysis dos not adress the point of analysis of the primary and secondary endpoint aligned with sample size justification. Analysis population is missing.

I recommended in my first review some aspects: The statistical analysis model has to be modified. The primary analysis is a

stratified t-Test (Age) mit MI aggregation for missing. The sensitivity analysis may

include a mixed model for Repeated measures (MMRM, Mallinckrod et al (2013),

"Recent Developments in the Prevention and Treatment of Missing Data")) using the

longitudinal observations. The latter should be used for the secondary outcomes as

well.

Reviewer #3: congratulation

your manuscript is acceptable............................................................................

7. PLOS authors have the option to publish the peer review history of their article (what does this mean?). If published, this will include your full peer review and any attached files.

Reviewer #1: No

Reviewer #3: **Yes: **Soheil Mansour Sohani

---

## [Author Response · Author response to Decision Letter 1]

29 Apr 2024

Reviewer #1: The presented description is not aligned with the specific study, i.e. what method to prove what! Statsistical analysis dos not adress the point of analysis of the primary and secondary endpoint aligned with sample size justification. Analysis population is missing.

I recommended in my first review some aspects: The statistical analysis model has to be modified. The primary analysis is a stratified t-Test (Age) mit MI aggregation for missing. The sensitivity analysis may include a mixed model for Repeated measures (MMRM, Mallinckrod et al (2013), "Recent Developments in the Prevention and Treatment of Missing Data")) using the longitudinal observations. The latter should be used for the secondary outcomes as well.

Thank you for the remark. However, we humbly disagree with the reviewer. Our study involves a repeated design, and therefore, the statistical approach should consider this appropriately. We refer to following; Studies have shown (eg. https://www.tandfonline.com/doi/full/10.1080/10543401003777995) that MMRM performed better than MI on longitudinal data. Additionally, the EMA guideline is also agnostic to the choice of the model (https://www.ema.europa.eu/en/documents/scientific-guideline/guideline-missing-data-confirmatory-clinical-trials_en.pdf). 

Finally, MI is also not needed MMRM (e.g. https://www.jclinepi.com/article/S0895-4356(13)00123-6/).

Due to the previous arguments, we would leave the analysis as it is. However, we leave it to the Editor to decide and will change it accordingly, if needed.

Reviewer #3: congratulation

your manuscript is acceptable.

---

## [Editor Report · Decision Letter 2]

7 May 2024

Effect of accelerated postoperative rehabilitation after tibial tubercle distalisation: a randomised controlled trial protocol

PONE-D-23-09669R2

Dear Dr. Timo Rahnel

We’re pleased to inform you that your manuscript has been judged scientifically suitable for publication and will be formally accepted for publication once it meets all outstanding technical requirements.

Kind regards,

Mehrnaz Kajbafvala, Ph.D

Academic Editor

PLOS ONE
---

## [Editor Report · Acceptance letter]

2 Jul 2024

PONE-D-23-09669R2 

PLOS ONE

Dear Dr. Rahnel, 

I'm pleased to inform you that your manuscript has been deemed suitable for publication in PLOS ONE. Congratulations! Your manuscript is now being handed over to our production team.

Kind regards, 

on behalf of

Dr. Mehrnaz Kajbafvala 

Academic Editor

PLOS ONE